

# Comparison of ultrasonic shear wave elastography, AngioPLUS planewave ultrasensitive imaging, and optimized high-resolution magnetic resonance imaging in evaluating carotid plaque stability

Shaoqin Zhang[1], Shuyan Jiang[1], Chunye Wang[2] and Chao Han[1]

[1] Department of Ultrasound, Yantaishan Hospital, Yantai, China
[2] Department of Imaging Division, Yantaishan Hospital, Yantai, China

## ABSTRACT

**Objective.** This study aimed to compare the efficiency of evaluating carotid plaque stability using ultrasonic shear wave elastography (SWE), AngioPLUS planewave ultrasensitive imaging (AP), and optimized high-resolution magnetic resonance imaging (MRI).

**Methods.** A total of 100 patients who underwent carotid endarterectomy at our hospital from October 2019 to August 2022 were enrolled. Based on the final clinical diagnosis, these patients were divided into vulnerable ($n = 62$) and stable ($n = 38$) plaque groups. All patients were examined using ultrasound SWE, AP, and optimized high-resolution MRI before surgery. The clinical data and ultrasound characteristics of patients of the two groups were compared. Considering the final clinical diagnosis as the gold standard, the sensitivity, specificity, positive predictive value (PPV), and negative predictive value (NPV) of SWE, AP, high-resolution MRI, and the final clinical diagnosis of vulnerable plaque were calculated. Pearson's correlation test was used to analyze the correlations of AP, SWE, and MRI results with the grading results of carotid artery stenosis.

**Results.** Statistically significant differences were noticed in terms of the history of smoking and coronary heart disease, plaque thickness, surface rules, calcified nodules, low echo area, and the degree of carotid artery stenosis between the two groups ($P < 0.05$). Considering the final clinical diagnosis as the gold standard, the sensitivity, specificity, PPV, and NPV of SWE-based detection of carotid artery vulnerability were 87.10% (54/62), 76.32% (29/38), 85.71% (54/63) and 78.38% (29/37), respectively, showing a general consistency with the final clinical results (Kappa = 0.637, $P < 0.05$). Considering the final clinical diagnosis as the gold standard, the sensitivity, specificity, PPV and NPV of AP-based detection of carotid artery vulnerability were 93.55% (58/62), 84.21% (32/38), 90.63% (58/64), and 88.89% (32/36), respectively, which agreed with the final clinical detection results (Kappa = 0.786, $P < 0.05$). Considering the final clinical diagnosis as the gold standard, the sensitivity, specificity, PPV and NPV of high-resolution MRI-based detection of carotid artery vulnerability were 88.71% (55/62), 78.95% (30/38), 87.30% (55/63), and 81.08% (30/37), respectively, showing consistency with the final clinical results (Kappa = 0.680, $P < 0.05$). AP, SWE, and

Corresponding author
Chao Han, handoctor2022@163.com

MRI results were positively correlated with the results of carotid artery stenosis grading ($P < 0.05$).

**Conclusion**. AP technology is a non-invasive, inexpensive, and highly sensitive method to evaluate the stability of carotid artery plaques. This method can dynamically display the flow of blood in new vessels of plaque in real time and provide a reference for clinical diagnosis and treatment.

## INTRODUCTION

Atherosclerosis is characterized by the formation of fibrous plaques in the walls of arteries; this subsequently leads to increased rigidity of the blood vessels, decreased blood flow, and increased systolic blood pressure. These changes are major cardiovascular and cerebrovascular events, such as cerebral infarction and acute myocardial infarction. Among blood vessel-related diseases, atherosclerosis is a major cause of high morbidity and mortality. Approximately 75% of patients with cerebral infarction have thrombosis due to atherosclerosis (*Huang et al., 2021*). Studies have shown that the formation of thrombosis is closely related to the nature of plaques, which can be classified depending on the histological features.

The main features of vulnerable plaques are intraplaque hemorrhage, neovascularization, thin fibrous cap, and ulceration, and these types of plaques have been associated with disease progression in patients with stroke (*Montanaro et al., 2021*). Rupturing or ulceration of a vulnerable plaque results in blood clotting, blocking the blood vessels and leading to cerebral infarction. During the plaque formation, the availability of sufficient amounts of oxygen to main homeostasis decreases substantially (severe hypoxia), stimulating neovascularization, and due to the absence of any connective tissue and basement membrane around the vessels of newly-formed plaques, these blood vessels can easily rupture, leading to bleeding and even thrombosis (*Karlöf et al., 2021*). Hypoxic conditions can promote angiogenesis by increasing the release of pro-angiogenic factors, such as vascular endothelial growth factor. Furthermore, the formation of new blood vessels can provide oxygen and nutrients to the plaques, allowing the growth and stability of these plaques. At the same time, neovascularization is mostly immature with weak walls that are prone to leakage; these weak vessels may lead to intraplaque hemorrhage and inflammatory reaction, increasing the risk of plaque rupture. Therefore, to reduce the risk of cerebral infarction and improve the prognosis of patients, improving the diagnostic accuracy of the properties of carotid plaques is of great significance. Ultrasound is a simple, safe, and noninvasive method that is also associated with providing consistent results. Among ultrasound-based technologies, shear wave elastography (SWE) can be used to quantify the elasticity of the internal tissue structure of plaques (*Li et al., 2021*). AngioPLUS planewave ultrasensitive imaging (AP) is a high-resolution technology and can quantitatively estimate the blood flow inside

micro-vessels and distinguish the direction of blood flow (*Ha et al., 2021*). SWE infers the hardness and elasticity of the tissue by applying mechanical waves to the plaque area and measuring the propagation velocity. The hardness of the plaque is related to its stability, and hence plaques with higher hardness may be more likely to rupture and form a thrombus. Therefore, SWE technology provides a non-invasive method to evaluate the stability of plaques. AP technology measures the velocity of blood flow and the volume and hemodynamic parameters of the plaque area. A few advanced techniques, such as automatic phase-correlated volume imaging, can provide more detailed microvascular images and distinguish the direction of blood flow. The analysis of blood flow direction can help to evaluate the pathological features of plaques, such as thrombosis and plaque instability. High-resolution magnetic resonance imaging (MRI) yields valuable results while evaluating the vulnerability of atherosclerotic plaques and plaque microstructure (*Han et al., 2020*). This study compared the performance of SWE, AP technology, and high-resolution MRI in assessing the stability and characteristics of carotid plaques. Furthermore, this study compared the consistency of different imaging techniques to clinically differentiate between the vulnerable plaque and stable plaque groups. This study aimed to provide valuable insights for the clinical evaluation of carotid plaques, supporting more accurate diagnoses and targeted treatment strategies for tissue lesions.

## MATERIALS & METHODS

### Subjects

This retrospective study included 100 patients who received carotid endarterectomy at our hospital from October 2019 to August 2022. This study was approved by the ethics committee of the Yantai Mountain Hospital, and this study followed the tenets of the Declaration of Helsinki. The requirement of obtaining the informed consent was waived. Patients were included in the study based on the following inclusion criteria: (1) patients with carotid atherosclerotic plaque thickness of >1.5 mm under conventional carotid ultrasound, (2) availability of complete clinical examination data, and (3) agreed to participate in this study and signed the informed consent. The exclusion criteria were: (1) patients contraindicated to ultrasound and MRI examinations; (2) patients with rheumatic heart disease, atrial fibrillation, arrhythmia, and other diseases; (3) patients with malignant tumor and liver and kidney insufficiency; (4) patients who had received treatment for carotid atherosclerosis before examination; and (5) patients with incomplete clinical data.

### Methods

#### Conventional ultrasound examination

A GE Logiq E9 color Doppler ultrasonic diagnostic instrument was used with the probe frequency set at 4–9 MHz. During the examination, the patients were instructed to lie in the supine position with the head tilted to the opposite side and the neck fully exposed. The carotid arteries, including both sides of the common carotid artery, the bifurcation of the carotid artery, the proximal end of the internal carotid artery, and the external carotid artery, were scanned using the probe, and the plaque was carefully observed. Details of plaque surface features, calcified nodules, and the low echoic area were recorded. Plaque

thickness, carotid intima-media thickness (IMT), and stenosis were also measured. The following were the conventional ultrasonic diagnosis criteria for unstable plaque: uneven echo, irregular plaque surface, active bleeding within the plaque, low internal echo, and incomplete fibrous cap. A normal IMT of 1.4 mm represented the carotid plaque. If bilateral common carotid artery and internal carotid artery intima were 0.9 mm $\leq$ IMT <1.3 mm, a thickening of the plaque was considered.

### SWE

The SWE examination was performed after a routine ultrasound examination. An appropriate nod was selected at the patient's clavicle to fix the probe, and the ultrasonic SWE mode was started. The range of Young's modulus (YM) was 1–100 kPa. The dynamic images were collected under the breath-holding state of the patient, and the images with uniform blue and red coverage were frozen. The embedded analysis system of the instrument was used to analyze the plaque hardness and trace, and Young's modulus values at different echo areas of the plaques were measured separately. The measurements were repeated three times for each plaque, and the mean was considered as the mean YM.

### AP technology

For this examination, radiologists referred to the vascular ultrasound guidelines formulated by the Sonographer Branch of the Chinese Medical Doctor Association. Transverse and longitudinal sections within the plaque were selected, and the scanning range included 1–1.5 cm above and below the carotid artery bifurcation. The whole plaque was continuously scanned, and continuous multi-section dynamic images were stored. The blood flow signals and abnormalities in and around the plaques of the bilateral common carotid artery, carotid ball, internal carotid artery, and external carotid artery were observed in real-time, and the peak systolic blood flow velocity (PSV) and end-diastolic blood flow velocity (EDV) of the above vessels were recorded. The following criteria were used to determine a stable plaque: smooth patches, intact fibrous caps, and homogeneous patches with moderate echo and strong echo. On the other hand, the following criteria were used to determine a vulnerable plaque: low-echo, moderate low-echo uneven plaque, irregular plaque morphology, incomplete fibrous caps, or blood flow signals in the plaque.

### High-resolution MRI

The American GE MR Discovery 750 superconducting scanner was used for scanning, and the high-resolution MRI of the carotid artery in the axial position was performed by a head-neck combined 20-channel coil and 4-channel coil for the carotid artery. During the examination, the patient's head and neck were kept in a fixed position, and the patient was instructed to remain silent with reduced swallowing. The scanning range was three cm above and below the carotid artery bifurcation, which was considered the center. Table 1 shows the sequence and main parameters of high-resolution MRI.

### Grouping method

The classification criteria of the American College of Cardiology standard (*Clarke et al., 2006*) was implemented. The vulnerable and stable plaques were considered to belong to

**Table 1 The sequence and main parameters of high-resolution MRI.**

| Parameter | T1WI-TRA | T1-SPC-COR-FS | T2-TSE-TRA | TSE-3D-T2WI-COR | 3D-TOF |
|---|---|---|---|---|---|
| TE | 9.2 | 11 | 65 | 120 | 4 |
| TR | 800 | 700 | 3500 | 1300 | 29 |
| FOV | 180 | 230 | 180 | 220 | 180 |
| FA | 180 | – | 138 | – | 18 |
| Matrix | $256 \times 256$ | $256 \times 256$ | $256 \times 256$ | $256 \times 256$ | $256 \times 256$ |
| Layer thickness | 2 | 0.9 | 2 | 0.9 | 1 |
| Number of layers | 20 | 192 | 20 | 56 | 40 |

types IV, V, and VI and types I, II, III, VII, and VIII, respectively. Based on the final clinical diagnosis, patients were divided into vulnerable ($n = 62$) and stable ($n = 38$) plaque groups.

## Observation targets

Considering the final clinical diagnosis as the gold standard, the sensitivity, specificity, positive predictive value (PPV), and negative predictive value (NPV) of SWE, AP technique, and high-resolution MRI were calculated; the final clinical diagnosis of vulnerable plaque was also calculated. The formulas used during the calculation were as follows: sensitivity = true positive number/(true positive number + false negative number) $\times 100\%$; specificity = true negative number/(true negative number + false positive number) $\times 100\%$; PPV = true positive number/(true positive number + false positive number) $\times 100\%$; and NPV = true negative number/(true negative number + false negative number) $\times 100\%$. To maintain the consistency of evaluation among observers, standardized operational procedures and evaluation indicators were developed and complied with. This included technical training, standardized image acquisition and analysis methods, consistent evaluation criteria, and classification systems.

## Statistical methods

The double-entry method was used to ensure that the collected data were true and reliable. SPSS22.0 software was used to analyze and process the data. Measurement data conforming to normal distribution were represented $\bar{X} \pm s$ and a $t$-test was used for performing comparisons. Counting data were represented as cases or %. The chi-square test was used to compare two groups, and $Z$-test was used to compare multiple groups. The correlation was evaluated using Pearson's correlation test. Sensitivity and specificity refer to the proportion of positive and negative results in the actual positive and negative results, respectively. PPV and NPV refer to the proportion of actual positive and negative results in the judgment of positive and negative results, respectively. Consistency was examined using the Kappa test; kappa values of $\geq 0.75$, $0.4 \leq$ Kappa $< 0.75$, and $< 0.4$ indicated good, average, and poor consistencies, respectively. The statistical significance was set at $P < 0.05$.
**Table 2  Comparison of general data between two groups.**

| Item | Vulnerable plaque group ($n = 62$) | Stable plaque group ($n = 38$) | $t/\chi^2$ value | $P$ value |
|---|---|---|---|---|
| Gender (male/female) | 43/19 | 24/14 | 0.409 | 0.523 |
| Age (years, $\bar{x} \pm s$) | 59.67 ± 3.54 | 59.31 ± 3.68 | 0.486 | 0.628 |
| BMI (kg/m$^2$, $\bar{x} \pm s$) | 25.88 ± 2.56 | 25.74 ± 2.78 | 0.257 | 0.798 |
| Smoking history (yes/no) | 53/9 | 26/12 | 4.135 | 0.042 |
| History of hypertension (yes/no) | 49/13 | 27/11 | 0.823 | 0.365 |
| History of hyperglycemia (yes/no) | 38/24 | 23/15 | 0.006 | 0.939 |
| History of hyperlipidemia (yes/no) | 44/18 | 23/15 | 1.162 | 0.281 |
| History of coronary heart disease (yes/no) | 41/21 | 14/24 | 8.165 | 0.004 |

**Table 3  Comparison of conventional ultrasonic features between two groups.**

| Group | Plaque thickness (mm, $\bar{x} \pm s$) | IMT (mm, $\bar{x} \pm s$) | Surface characteristics (regular/irregular) | Calcified nodules (yes/no) | Low echo area (yes/no) | Degree of carotid stenosis (mild/moderate/severe/occlusive) |
|---|---|---|---|---|---|---|
| Vulnerable plaque group ($n = 62$) | 3.23 ± 0.63 | 1.37 ± 0.38 | 21/41 | 22/40 | 34/28 | 10/21/27/4 |
| Stable plaque group ($n = 38$) | 2.06 ± 0.77 | 1.33 ± 0.41 | 29/9 | 30/8 | 11/27 | 26/11/1/0 |
| Inspection value | 8.276 | 0.496 | 16.981 | 17.831 | 6.381 | −5.828 |
| $P$ value | 0.000 | 0.621 | <0.001 | <0.001 | 0.012 | 0.000 |

## RESULTS

### Comparison of general data between two groups

No significant differences were observed in sex, age, body mass index (BMI), and history of hypertension, hyperglycemia, and hyperlipidemia between the two groups ($P > 0.05$); however significant differences were observed in terms of history of smoking and coronary heart disease ($P < 0.05$) (Table 2).

### Comparison of conventional ultrasonic features between two groups

No significant difference was observed in IMT between the two groups ($P > 0.05$). However, significant differences were noticed between the two groups in terms of plaque thickness, surface regularity, calcified nodules, low echo area, and carotid artery stenosis ($P < 0.05$) (Table 3).

### Consistency between SWE and clinical diagnosis in assessing plaque vulnerability

Considering the final clinical diagnosis as the gold standard, the sensitivity, specificity, PPV, and NPV of SWE in assessing carotid artery vulnerability were 87.10% (54/62), 76.32% (29/38), 85.71% (54/63), and 78.38% (29/37), respectively. These results were in agreement with the final clinical results (Kappa = 0.637, $P < 0.05$, Table 4).

**Table 4 Consistency between SWE and clinical diagnosis in assessing plaque vulnerability.**

| SWE | Clinical diagnosis | | Total |
| --- | --- | --- | --- |
| | Vulnerable plaque | Stable plaque | |
| Vulnerable Plaque | 54 | 9 | 63 |
| Stable plaque | 8 | 29 | 37 |
| Total | 62 | 38 | 100 |

**Table 5 Consistency between AP and final clinical diagnosis in assessing plaque vulnerability.**

| AP | Clinical diagnosis | | Total |
| --- | --- | --- | --- |
| | Vulnerable plaque | Stable plaque | |
| Vulnerable Plaque | 58 | 6 | 64 |
| Stable plaque | 4 | 32 | 36 |
| Total | 62 | 38 | 100 |

**Table 6 Consistency between high-resolution MRI and final clinical diagnosis in assessing plaque vulnerability.**

| High-resolution MRI | Clinical diagnosis | | Total |
| --- | --- | --- | --- |
| | Vulnerable plaque | Vulnerable Plaque | |
| Vulnerable Plaque | 55 | 8 | 63 |
| Stable plaque | 7 | 30 | 37 |
| Total | 62 | 38 | 100 |

## Consistency between AP and final clinical diagnosis in assessing plaque vulnerability

Considering the final clinical diagnosis as the gold standard, the sensitivity, specificity, PPV and NPV of AP in assessing carotid artery vulnerability were 93.55% (58/62), 84.21% (32/38), 90.63% (58/64), and 88.89% (32/36), respectively. These results agreed with the final clinical results (Kappa = 0.786, $P < 0.05$, Table 5).

## Consistency between high-resolution MRI and final clinical diagnosis in assessing plaque vulnerability

Considering the final clinical diagnosis as the gold standard, the sensitivity, specificity, PPV, and NPV of high-resolution MRI in assessing carotid artery vulnerability were 88.71% (55/62), 78.95% (30/38), 87.30% (55/63), and 81.08% (30/37), respectively. These results agreed with the final clinical results (Kappa = 0.680, $P < 0.05$, Table 6).

## Correlation between AP/SWE/MRI examination results and grading results of carotid artery stenosis

The results of the AP technique, SWE, and MRI were positively correlated with the grading results of carotid artery stenosis ($P < 0.05$, Table 7).

**Table 7 Correlation between AP/SWE/MRI examination results and grading results of carotid artery stenosis.**

| Correlation | AP | SWE | MRI |
|---|---|---|---|
| $r$ | 0.743 | 0.624 | 0.669 |
| $P$ | <0.05 | <0.05 | <0.05 |

## DISCUSSION

This study evaluated the stability of carotid plaques in a cohort of patients with stroke. Patients of the vulnerable and stable groups had statistically significant differences in terms of smoking history, coronary heart disease history, plaque thickness, surface rules, calcified nodules, low echo area, and the degree of carotid artery stenosis ($P < 0.05$). The results of SWE, AP, and MRI demonstrated average (Kappa =0.637, $P < 0.05$), superior (Kappa =0.786, $P < 0.05$), and average (Kappa =0.680, $P < 0.05$) consistency with the final clinical results, respectively. The results of SWE, AP, and MRI were positively correlated with the grading results of carotid artery stenosis ($P < 0.05$).

At present, carotid atherosclerotic plaque is one of the common diseases affecting human vascular health. Vulnerable plaques are prone to rupture and cause thrombosis; these plaques may rapidly develop into liability lesions, which are closely related to ischemic cerebrovascular disease (*Wang et al., 2022b*). Vulnerable plaques can promote the development of atherosclerotic lesions, and they can even induce intra-plaque bleeding and plaque rupture mainly due to the irregular surface of plaques or formation of ulcerative plaques, resulting in small blood vorticity, platelet aggregation, and the formation of blood clots that break off easily (*Biswas et al., 2021*). Hence, accurate assessment of the stability of carotid atherosclerotic plaque is clinically crucial. Ultrasound, computed radiography (CR), MRI, and other imaging methods are widely used in clinical evaluation of the stability of carotid atherosclerotic plaque. The spatial resolution of CR and MRI is relatively low, which may not be able to accurately display the small structure and subtle changes of the plaque and cannot provide detailed information on the composition of the plaque tissue. Conventional ultrasound can observe carotid MT, plaque thickness, and carotid artery stenosis. The present study observed significant differences in plaque thickness, surface regularity, calcified nodules, low echo area, and carotid artery stenosis between the two groups. Furthermore, IMT did not show any significant difference between the two groups, which may be related to the small sample size and interference caused by human factors. The result of conventional ultrasound is majorly affected by the subjective judgment of the examining physician. Furthermore, the capacity of conventional ultrasound to identify the components within the plaques is poor (*Huang et al., 2022*), revealing certain limitations in the judgment of plaque stability. SWE is an imaging technique used to quantify plaque elasticity. This method uses shear waves generated by ultrasound to evaluate the hardness or elastic properties of tissues. In SWE, the transverse shear wave generated by the ultrasonic beam propagates in the tissue and is received and measured by the ultrasonic probe. According to the propagation velocity of the shear wave, the elastic modulus of the tissue can be calculated, and subsequently, the hardness of the plaque

can be adjudged. AP technology is used to measure microvascular blood flow. It uses the acoustic radiation force of ultrasound to stimulate small vibrations in the tissue and then uses a synchronous ultrasound probe to detect and quantify the direction and velocity of blood flow. By analyzing the nature of the small vibrations caused by acoustic radiation force, the direction of blood flow can be identified, and the relevant blood flow parameters, such as velocity and volume, can be calculated to help doctors accurately assess plaque stability, blood flow supply, and cardiovascular risk.

The principle of SWE is to apply the directional force to the tissue to generate shear waves for acoustic wave collection. The hardness of the tissue can be determined by detecting the propagation speed of acoustic waves in the tissue. The thin fibrous caps, lipid cores, and new blood vessels inside the plaque affect the elasticity of the plaque itself (*Zhang et al., 2021*). As a basic physical unit that can be used to quantify the hardness of patches, YM can reflect the composition of plaques, and then serve as an indicator to evaluate the vulnerability of plaques. SWE technology can provide quantitative indicators for evaluating the stiffness and softness of carotid plaques in a wide range and is a supplement to the traditional ultrasonic examination of atherosclerotic plaques (*Goudot et al., 2022*). Currently, SWE technology is mainly used for clinical diagnosis of breast, liver, and other malignant tumors (*Ko et al., 2020*; *Yoo et al., 2020*). The plaque hardness can be determined by the YM-calculated SWE velocity value; the harder the plaque, the faster the propagation velocity (*Celletti et al., 2021*). AP technology is a type of Doppler ultrasound technology that detects low-speed blood flow in micro-vessels through the principle of unfocused or planewave and three-dimensional wall filtering. It can analyze tissue movement from the aspects of time, space, and amplitude, to effectively distinguish blood flow and tissue information (*Jung et al., 2018*; *Wang et al., 2022a*). MRI imaging technology is characterized by multi-plane and multi-parameter and has a higher resolution for soft tissue. Compared with traditional MRI scanning technology, high-resolution MRI has a higher image resolution and can dynamically observe nourishing blood vessels in plaques and active inflammation in vulnerable plaques (*Sun et al., 2020*). It has been found (*Zheng et al., 2022*) that the 3D-TOF sequence can effectively distinguish the status of plaque fibrous cap, and DP-TSE technology can dynamically observe the changes of new vessels and inflammatory cell infiltration in carotid plaque without the use of contrast agents to accurately assess the vulnerability of plaque.

By calculating the sensitivity, specificity, PPV, and NPV of SWE, AP technology, high-resolution MRI, and the final clinical detection of vulnerable plaque, this study analyzed the value of SWE, AP, and high-resolution MRI in evaluating the stability of carotid plaque. With the final clinical diagnosis as the gold standard, the sensitivity, specificity, PPV and NPV of SWE in detecting carotid artery vulnerability were 87.10% (54/62), 76.32% (29/38), 85.71% (54/63) and 78.38% (29/37), respectively, in a general agreement with the final clinical results (Kappa =0.637); these four indicators of AP were 93.55% (58/62), 84.21% (32/38), 90.63% (58/64) and 88.89% (32/36), respectively, in a good agreement with the final clinical results (Kappa = 0.786); and the values for high-resolution MRI in assessing carotid artery vulnerability were 88.71% (55/62), 78.95% (30/38), 87.30% (55/63) and 81.08% (30/37), respectively, in a general agreement with
the final clinical results (Kappa = 0.680). The results of SWE and high-resolution MRI in assessing the stability and characteristics of carotid plaques were similar and inferior to that of AP, which may be attributed to the fact that ultrasonic AP technology can display the low-velocity blood flow signals of new vessels in plaques more sensitively and thus determine the abnormal morphology of the new vessels in plaques. SWE is less dependent on the sonographer's experience than traditional ultrasound. Relevant literature has shown (*Marlevi et al., 2020*) that the feasibility of applying SWE in carotid plaque examination can be determined by evaluating the reproducibility of the experimental vascular model. The study showed that SWE has feasibility and good reproducibility in quantifying YM of the vascular wall and carotid plaque model, and can help to identify the risk of unstable carotid plaque even in the presence of pulsating arterial tissue motion. Meanwhile, by comparing the "gold standard" histopathological results of plaque vulnerability assessment (*Zamani et al., 2020*), it was found that YM significantly decreased in the identification of plaques with bleeding, thrombosis, and a large number of foam cells by SWE, suggesting that SWE has a good value in the identification of features related to unstable plaques such as bleeding within plaques and the presence of lipid cores within plaques. Moreover, the technical equipment has disadvantages such as high cost, long examination time, and many contraindications, which greatly limits its wide clinical application. High-resolution MRI may have some contraindications, including patients with cardiac pacemakers, internal hearing aids, nerve stimulators, or other implants. These devices may be disturbed by a magnetic field, which may cause damage to the device or pose a risk to the patient. In addition, for some patients, there may be a fear of narrow space or an inability to tolerate long-term scanning.

Because high-resolution MRI requires more detailed image information, its examination time is usually longer than traditional MRI. This may be a problem for some patients because of the need to keep the position still for a limited time. In addition, long-term examination may also cause inconvenience and discomfort to patients, especially for patients with movement disorders or unsuitable for long-term lying.

This study is not sequential. The study participants had high-risk plaques; however, no ischemic stroke events occurred during the study period. Therefore, to further verify the value of high-risk plaques in early warning of ischemic stroke, long-term follow-up of multiple samples is needed to observe the changes in carotid plaques. This study is also limited by the lack of sample size, which will be further focused and considered in future work.

## CONCLUSIONS

Ultrasound AP technology demonstrated superior results that were consistent with the final clinical diagnosis regarding the stability of carotid plaque. In contrast, the results of SWE and high-resolution MRI had an average level of consistency with the final diagnosis. Ultrasonic AP technology can dynamically display the flow of blood in new vessels of plaques in real time and provide a reference for clinical diagnosis and treatment. The findings of this study may improve the diagnostic accuracy of carotid plaque characteristics, thereby effectively reducing the risk of cerebral infarction and improving the prognosis of patients.

## ACKNOWLEDGEMENTS

I would like to express my gratitude to all those who helped me during the writing of this thesis. I also acknowledge the help of my colleague, Yanfang Cui, for providing suggestions during my academic studies.

### Funding

This work was supported by the Project of Yantai Science and Technology Bureau (No. 2022YD063) and the Shandong Province Traditional Chinese Medicine Technology Development Plan Project (No. 2019-0686). The funders had no role in study design, data collection and analysis, decision to publish, or preparation of the manuscript.

### Grant Disclosures

The following grant information was disclosed by the authors:
Project of Yantai Science and Technology Bureau: 2022YD063.
Shandong Province Traditional Chinese Medicine Technology Development Plan Project: 2019-0686.

### Competing Interests

The authors declare there are no competing interests.

### Author Contributions

- Shaoqin Zhang conceived and designed the experiments, performed the experiments, authored or reviewed drafts of the article, and approved the final draft.
- Shuyan Jiang conceived and designed the experiments, performed the experiments, analyzed the data, prepared figures and/or tables, and approved the final draft.
- Chunye Wang analyzed the data, prepared figures and/or tables, authored or reviewed drafts of the article, and approved the final draft.
- Chao Han performed the experiments, analyzed the data, prepared figures and/or tables, authored or reviewed drafts of the article, and approved the final draft.

### Human Ethics

The following information was supplied relating to ethical approvals (i.e., approving body and any reference numbers):

All samples obtained in this study were approved by the ethics committee of the Yantai Mountain Hospital and abided by the ethical guidelines of the Declaration of Helsinki.

### Data Availability

The raw data are available in the Supplementary File.

### Supplemental Information

Supplemental information for this article can be found online at http://dx.doi.org/10.7717/peerj.16150#supplemental-information.

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
