# Peer review of "Comparison of ultrasonic shear wave elastography, AngioPLUS planewave ultrasensitive imaging, and optimized high-resolution magnetic resonance imaging in evaluating carotid plaque stability"

_PeerJ, doi:10.7717/peerj.16150_

## Round 0.1 · original submission · Minor Revisions

All three reviewers provided constructive and helpful comments. I believe that revising and refining your research in accordance with these suggestions and comments will go a long way in improving the quality of the paper. I note that the paper has language problems that need to go into careful correction of the manuscript. For example, a or the is missing before the noun. Also, line 37: [Ultrasonic AP technology has the advantages of non-invasive] should be revised to [Ultrasonic AP technology has the advantages of being non-invasive]. Please complete the revision as soon as possible and resubmit the paper.

**Language Note:** The Academic Editor has identified that the English language must be improved. PeerJ can provide language editing services - please contact us at copyediting@peerj.com for pricing (be sure to provide your manuscript number and title). Alternatively, you should make your own arrangements to improve the language quality and provide details in your response letter. – PeerJ Staff

Reviewer 1 ·

Basic reporting

1. The introduction may benefit from clearly stating the specific objective of the study, such as "This study aims to evaluate and compare the diagnostic value of ultrasonic SWE, AP technology, and optimized high-resolution MRI in assessing the stability of carotid plaques."
2. It would greatly enhance the introduction to include a concise definition and highlight the significance of atherosclerosis. For instance, including a description such as "Atherosclerosis is a chronic inflammatory disease characterized by the build-up of plaques in the arterial walls. It is a major cause of various vascular diseases, including cerebral infarction."
3. It could be beneficial to expand on the correlation between vulnerable plaques and disease progression within the introduction. The statement about vulnerable plaques being correlated with disease progression in stroke patients lacks specific details. It would be valuable to provide more information on the evidence supporting this correlation.
4. Avoid repetitive phrases: The paragraph includes repetitive phrases such as "diagnostic accuracy" and "evaluating the stability." Avoid redundant use of similar terms to improve clarity and readability. For example, instead of "evaluating the stability of carotid plaques," consider "assessing the stability and characteristics of carotid plaques."
5. Proofread the paragraph for grammar, sentence structure, and phrasing. Ensure that each sentence is clear and concise.
6. It would be helpful to define the terms "vulnerable plaque" and "stable plaque" for readers who may not be familiar with these terms.

Experimental design

7. Please state the advantages of shear wave elastography (SWE) and Angio plus TM plane wave ultrasensitive imaging (AP). Describe how SWE quantifies plaque elasticity and how AP technology measures micro-vessel blood flow and distinguishes blood flow direction. Emphasize how these techniques contribute to evaluating plaque properties and why they are important in clinical settings.
8. The statement that "Ultrasound, computed radiography (CR), MRI and other imaging methods are widely used in clinical evaluation of the stability of carotid atherosclerotic plaque" could benefit from specific examples of how each imaging method is used and their respective strengths and limitations.
9. The sentence "This suggested that there may be some errors in the conventional ultrasonic measurement of relatively thin IMT, While the measurement of relatively thick IMT is more accurate" is unclear and contradictory. It needs clarification and justification. If the results suggest errors in the measurement of thin IMT, explain the possible reasons and limitations of ultrasound in measuring thin IMT accurately. If the measurement of thick IMT is claimed to be more accurate, provide evidence or studies supporting this claim.
10. The paragraph describing the principle of SWE would benefit from a clearer explanation of how the technique assesses plaque composition and measures plaque stiffness.
11. The limitations of high-resolution MRI should be discussed in more detail, including the specific contraindications, cost factors, and the impact of long examination times on patient convenience and accessibility.

Validity of the findings

12. Add a sentence summarizing the anticipated contributions or key findings of the study. For example, "The findings of this study have the potential to improve the diagnostic accuracy of carotid plaque properties, leading to effective risk reduction of cerebral infarction and improved prognosis for patients."
13. The results section lacks an overall introduction or summary. Consider adding a brief introductory paragraph that provides an overview of the main findings before diving into specific comparisons and correlations.
14. Justify the statement that AP technology has "the advantages of non-invasiveness, inexpensiveness, and high sensitivity." Provide supporting references or evidence for these claims.
15. The sentence "Different sequences of high-resolution MRI have a good recognition effect on various components of plaque, and the ordinary consistency with the final clinical detection results may be related to the sample size" is unclear and lacks clarity. Please clarify the sentence regarding the recognition effect of high-resolution MRI sequences on different plaque components. Explain how different sequences are used and highlight their ability to identify specific plaque features. Additionally, clarify the role of sample size in the observed consistency between high-resolution MRI and final clinical detection results.
16. Discuss the potential implications of the study findings on clinical practice and future research in the field of carotid plaque stability assessment. Highlight any novel insights or strategies that may arise from the study and suggest possible areas for further investigation. This will provide a broader perspective and show the significance of the research in the context of clinical applications.

Additional comments

17. Elaborate on the mechanisms underlying the link between hypoxia, neovascularization, and plaque rupture. Provide specific references to support these statements.
18. Include references or specific examples that demonstrate the efficacy of high-resolution MRI in evaluating the vulnerability of atherosclerotic plaques and assessing plaque microstructure. This will further reinforce the importance of high-resolution MRI in clinical evaluations.
19. Add an introductory paragraph summarizing the main findings of the study, such as the comparisons between vulnerable plaque and stable plaque groups and the consistency between different imaging techniques and clinical diagnosis.

Reviewer 2 ·

Basic reporting

1) Provide a brief description of the software or analysis tools used to calculate the sensitivity, specificity, PPV, and NPV of each imaging technique. Explain if any adjustments or corrections were applied during the calculation process.
2) Review the paper for grammatical and typographical errors and revise as necessary for clarity and readability.

Experimental design

3) Add a sentence mentioning the reliability assessment of the measurements taken during the conventional ultrasound examination. Since these measurements might be subject to variability, explain any measures taken to ensure consistency and reliability, such as inter-observer reliability assessment or quality control protocols.
4) Specify the statistical methods used for analyzing the agreement or consistency between different imaging techniques (e.g., kappa test). Describe how the agreement was interpreted and what levels of consistency indicate good, average, or poor agreement.
5) Consider including confidence intervals or p-values for the correlations between AP, SWE, MRI, and carotid artery stenosis. This will provide information on the statistical significance and precision of the observed correlations.
6) Expand on the grading results of carotid artery stenosis. Provide a brief description of the grading system used, including specific criteria or thresholds applied to categorize the severity of stenosis.
7) If available, provide inter-observer or intra-observer reliability measures for the AP, SWE, and MRI examinations. This will address the variability of the measurements taken and increase confidence in the results.

Validity of the findings

8) Consider rephrasing the sentence about providing a reference for clinical evaluation: The sentence "with the aim of providing a reference for clinical evaluation of tissue lesions" could be rephrased to clarify the intended outcome of the study. Revision suggestion: "This study aims to provide valuable insights for the clinical evaluation of carotid plaques, supporting more accurate diagnoses and targeted treatment strategies for tissue lesions."
9) Include the sample size and demographic characteristics (e.g., age, gender distribution) of the vulnerable plaque group and stable plaque group. This will provide important context for the interpretation of the study findings.
10) Clarify the measurement units used for plaque thickness, IMT, low echo area, and carotid artery stenosis. This will ensure clarity and reproducibility of the results.
11) Clarify the interpretation of statistical measures such as sensitivity, specificity, PPV, NPV, and Kappa. Explain how these measures indicate the accuracy or agreement between the imaging techniques and clinical diagnosis.
12) Provide the confidence intervals (if available) for sensitivity, specificity, PPV, NPV, and Kappa values. This additional information will help to assess the precision and reliability of the estimated measures.

Additional comments

13) Include a sentence explaining the rationale for adopting the classification criteria of the American College of Cardiology to categorize plaques as stable or vulnerable. State how these criteria are widely accepted in the field and provide references if necessary.
14) Add more context to the correlation analysis results between AP, SWE, MRI, and carotid artery stenosis. Describe the strength and direction of the correlations observed.
15) If there are any limitations or potential biases in the study methodology or data analysis, explicitly acknowledge and discuss them. This will help readers evaluate the reliability and generalizability of the findings.

Reviewer 3 ·

Basic reporting

1. Improve the flow and organization. The manuscript could benefit from improved flow and organization. Consider reorganizing the sentences to follow a logical progression of ideas and concepts.
2. Clarify the scope of the study: It's important to clearly define the scope of the study to avoid any ambiguity. State whether the study is conducted on a specific population or if it focuses on a particular aspect of carotid plaques. Maybe this manuscript could include a sentence specifying the scope of the study. For example, "This study primarily focuses on evaluating the stability of carotid plaques in a cohort of stroke patients, considering their diagnostic value in assessing plaque characteristics."
3. Add a sentence clarifying the statistical method used to analyze the agreement or consistency between different imaging techniques (e.g., kappa test). Include a sentence explaining the interpretation of the agreement levels, such as good, average, or poor consistency, using appropriate thresholds.

Experimental design

Good.

Validity of the findings

4. Provide information on the qualifications and experience of the individuals performing the imaging examinations to ensure consistency and reliability of the results. Whether a same examiner performed all examinations for all subjects.
5. Include inter-observer or intra-observer reliability measures (e.g., Cohen's kappa, intraclass correlation coefficients) for the AP, SWE, and MRI examinations. This will indicate the consistency or reproducibility of the measurements and enhance the reliability of the study findings.

Additional comments

6. Include the rationale for choosing the specific number of patients (100) for the study. Justify why this sample size is appropriate to achieve the study objectives.
7. Specify the settings and parameters used during each imaging technique (ultrasound, SWE, AP technology, high-resolution MRI). Provide details such as image resolution, imaging sequences, and specific settings configured for each technique.
8. Include a paragraph outlining the process of data collection and handling. Describe any quality control measures undertaken, such as data validation, double-checking, or inter-rater reliability assessments to enhance the accuracy and reliability of the collected data.
9. Consider providing additional information about the characteristics of the patient population, such as age, gender distribution, and relevant clinical factors. This will help to better understand the demographic and clinical context of the study.
10. Include the rationale for selecting the specific criteria for inclusion and exclusion of patients. Explain the scientific basis for choosing these criteria and how they contribute to achieving the study objectives.

---

## Round 0.2 · Minor Revisions

Unfortunately, my concerns were not adequately addressed. Suitable modifications are needed to further enhance the quality of that study.

For instance:
1. Line 37: [Ultrasonic AP technology has the advantages of non-invasive] should be revised to [Ultrasonic AP technology has the advantages of being non-invasive].
2. Line 35: [AP, SWE, MRI results] should be [AP, SWE, and MRI results].
3. Line 161: [teseted] should be [tested].
4. Line 207: [showing] should be [showed].
5. Line 302: [a ordinary consistency] should be [an ordinary consistency].

In addition to these issues. the revised article still contains a large number of grammatical errors and spelling errors, please provide proof of editing from a professional English editing service.

**Language Note:** The Academic Editor has identified that the English language must be improved. PeerJ can provide language editing services - please contact us at copyediting@peerj.com for pricing (be sure to provide your manuscript number and title). Alternatively, you should make your own arrangements to improve the language quality and provide details in your response letter. – PeerJ Staff

---

## Round 0.3 · accepted · Accept

Three experts have approved this revised article for publication. I also independently evaluated the article and I was satisfied with the responses and revisions made by the authors. With the necessary revisions and improvements, the quality of this paper has been significantly improved. I believe that this revised manuscript is ready to be considered for publication in this journal.

Reviewer 1 ·

Basic reporting

no comment

Experimental design

no comment

Validity of the findings

no comment

Additional comments

The author has made good revisions according to my suggestions, and I have no further review comments.

Reviewer 2 ·

Basic reporting

No

Experimental design

No

Validity of the findings

No

Additional comments

The author has made good revisions to this article and carefully revised my opinions. I think this article has met the publishing standards.

Reviewer 3 ·

Basic reporting

Good!

Experimental design

Good!

Validity of the findings

Good!

Additional comments

Your revised manuscript is acceptable for publication.